# Mortality, loss to follow-up and advanced HIV disease following virologic success in West African HIV-2 patients

**Jean J. Koffi**[1,2*], **Simon P. Boni**[1,3], **Lionèle Mba**[1,4], **Alexandra Bitty-Anderson**[1,5], **Annick G. Tchabert**[1], **Frank Y. Toure**[1], **Patrick A. Coffie**[1,2,6], **Andre Inwoley**[4,6], **Serge Paul Eholie**[1,2,6], **Didier K. Ekouevi**[1,5,7], **Boris Tchounga**[1,8]

**1** Programme PACCI site ANRS de Côte d'Ivoire, Abidjan, Côte d'Ivoire, **2** Service des Maladies Infectieuses et Tropicales, CHU de Treichville, Abidjan, Côte d'Ivoire, **3** Programme de lutte contre le cancer, Abidjan, Côte d'Ivoire, **4** Research and Diagnosis Center for AIDS and other infectious diseases (CeDReS), CHU (University Hospital) of Treichville, Abidjan, Côte d'Ivoire, **5** Centre Inserm 1219 & Institut de Santé Publique d'épidémiologie et de développement, Université de Bordeaux, Bordeaux, France, **6** Université Félix Houphouët-Boigny, Abidjan, Côte d'Ivoire, **7** Département de santé Publique, Faculté des Sciences de la santé, Université de Lomé, Lomé, Togo, **8** Elizabeth Glazer Pediatric A.I.D.S. Foundation, Yaoundé, Cameroon

* jean-jacques.koffi@pac-ci.org

## Abstract

### Background

People living with HIV-2 are mainly found in West Africa and their identification and treatment have been impaired by diagnostic challenges and availability of effective antiretroviral treatment (ART). With the roll out of first line dolutegravir (DTG)-based regimen, the situation may have improved, emphasizing the need for data on long-term treatment outcomes and advanced HIV disease among ART-experienced people living with HIV-2.

### Method

A prospective cohort was initiated in 2012 in Côte d'Ivoire and Burkina Faso. All adult patients from Côte d'Ivoire with an undetectable viral load, were included and followed up. HIV-2 viral load and CD4 counts were done during the routine follow-up visits and a detailed clinical assessment was done during the last follow up visit of the year 2018 corresponding to the censor date of the cohort. Outcomes were described as follow: in care (known alive and present during the last ART follow up visit), loss to follow-up (absent for more than 90 days and not reported dead), and dead (reported dead with a date of event). Advanced HIV disease followed WHO definition and virologic failure was define as viral load >50 copies/mm$^3$. The Kaplan-Meier curve was used to estimate mortality and Loss to follow-up probability.

### Results

Among the 108 HIV-2 patients in virologic success in 2012, 95 agreed to participate and were enrolled in the "success cohort". Their median age was 53 [47–60] years and all of

**Data availability statement:** The data is available on figshare at the following address: https://figshare.com/s/a8b9fa7d5ba8a5968235

**Funding:** The author(s) received no specific funding for this work.

**Competing interests:** The authors have declared that no competing interests exist.

them were receiving boosted-lopinavir-based ART regimen. Of the 95 participants, 65 (68.4%) remained in care, 20 (21.1%) were loss to follow-up and 10 (10.5%) were reported dead. The survival analysis retrieved a decreasing probability of remaining alive and in care over the time, moving from 90% to 80.7% and to 73.0% after 24, 48 and 72 months respectively. Overall, 36 (37.9%) patients presented with advanced HIV disease at their last visit, higher among those dead/ loss to follow-up compared to those remaining in care (60.0% vs 27.7%; p-0.003).

## Conclusion

High advanced HIV disease rate was found in HIV-2 patients, six years after an initial virologic success. This emphasizes the need to enable the one-stop-shop model that allow an early management of opportunistic infections while integrating non-communicable diseases services in HIV-2 care.

## Introduction

HIV-2 is endemic in West Africa and in countries with historical ties to this part of the world [1]. Compared with HIV-1 infection, HIV-2 is characterized by a longer asymptomatic stage, lower plasma viral load, decreased mortality rate due to AIDS, lower rates of sexual transmission [2–5]. However, without appropriate care and treatment, a significant proportion of people living with HIV-2 (PLHIV-2) progress to AIDS disease [2,6], and experience AIDS-related morbidity and mortality [7].

HIV-2 is also characterized by an intrinsic resistance to non-nucleoside reverse transcriptase inhibitors (NNRTI) and the suboptimal treatment response to some protease inhibitors (PI), as mainly three PI have potent activity against HIV-2: lopinavir, saquinavir and darunavir [8–11]. Recent data indicate that integrase strand transfer inhibitor (INSTI) class is safe and effective against HIV-2 [12,13]. In addition, the 2019 WHO guidelines recommend dolutegravir (DTG) combined with a nucleoside reverse transcriptase inhibitor (NRTI) as first-line treatment for people starting antiretroviral therapy (ART), including those living with HIV-2. [14].

In addition to the long-term management of ART regimen, the monitoring of treatment response is also challenging for PLHIV-2. Data from the Natural history of HIV-2 infection in adult patients living in France Cohort Study (ANRS CO5 HIV-2) showed lower than expected CD4 cell recovery in HIV-2 treated patients, despite the high virologic suppression rate [15]. Viral load assay for HIV-2 was recently approved and is still on the process for routine utilization as per the national guidelines [16].

As the treatment options for the management of PLHIV-2 are improving with the roll out of DTG, limited data are available on the long-term treatment outcome among PLHIV-2 in West Africa. This study aimed at describing mortality, loss to follow-up (LTFU), advanced HIV disease and virologic suppression in PLHIV-2 in West Africa.

## Methods

### Study design, population and settings

A cohort of PLHIV-2 identified as monoinfected and followed up in HIV clinics was initiated in January 2012. These HIV clinics were all participating in the International Epidemiological Database to Evaluate AIDS (IeDEA) West Africa. Participants were enrolled based on the

results of a cross-sectional survey assessing virologic failure among treatment experienced HIV-2 patients [17]. All adult patients from Côte d'Ivoire with an undetectable viral load in the 2012-virologic survey were enrolled in the "success cohort" and followed up as per the national PLHIV treatment guidelines [18].

## Follow-up procedures

As per the national guidelines for PLHIV, clinically and virologically stable patients should have a systematic quarterly visit at HIV clinic for ART drug refill and medical check. In case of missed appointment, a phone-base tracking procedure has been taken place to bring the patient back to care, and if not successful, home visits were organized for the patients who agreed and provided localization of their house at enrolment. Blood sample was collected every six months from each patient to perform CD4 count measurements. The last visit for all participants took place between October and December 2018. To ensure consistency in analysis, they were assessed by the same clinician who conducted a thorough clinical examination, updated the disease's clinical stage (if applicable), and finally collected blood samples for viral load testing. Those who missed their last appointment or who were known LTFU in the facility were tracked by phone to ascertain their vital status (death, alive) and their treatment situation.

## Data collection

A structured questionnaire allowed data to be extracted from patients' medical records, and additional data were obtained through face-to-face interviews with patients or abstracted from the facility's database. Data collected included baseline socio-demographic (age, gender, education status, marital status), HIV follow-up characteristics (duration of infection, ART, WHO clinical stage), immunological and virological response (history of CD4 cells, viral load). Follow-up data included clinical response (history of clinical stage), immunological response (history of CD4 count), virological response (history of viral load measurement) and follow-up outcomes including LTFU, and death.

## Outcomes variables

The main outcome variables observed in this analysis were death, LTFU and advanced HIV disease. LTFU was defined as being absent for more than 90 days, not reported death and not transferred out and unable to reach after a completed tracking procedure. Death was defined as being reported dead in facility register or documented as dead with a death certificate. Advanced HIV disease was defined according to WHO guideline definition: CD4 count $< 200$ cell/mm$^3$ or WHO clinical stage 3 or 4. Clinical stage was defined using WHO recommended staging process and was evaluated on a quarterly basis at each follow-up visit. Based on their follow up status, participants were categorized as in care (still in the follow-up and receiving ART, including being transferred out), dead and LTFU. The last two were subsequently combined in one category in the analyses, as they both constitute a negative outcome and it is always challenging to know the proportion of deaths among LTFU [19].

## Statistical analysis

Data analysis was performed using STATA® version 12.0, Stata Corp, College Station, Texas USA. Quantitative variables were described as medians with their interquartile range (IQR), and categorical variables were presented as proportions. Chi-square (X2) or Fisher's exact tests were used to compare proportions. The Kaplan-Meier survival curve was used to estimate mortality and LTFU probability among the participants. Variables were statistically significant when the p value $< 0.05$.

### Ethical considerations

The protocol of this study was approved by the national Institutional Review Board of Côte d'Ivoire, the "Comité National d'Ethique des Sciences de la Vie et de la Santé" (approval number *144-18/MSHP/CNESVS-km*). Prior to the enrolment in the cohort, detailed information about the study was given to each study participants and a signed informed consent was obtained. Data of the participants were anonymized during the abstraction and no identifiable data was collected or entered in the database. The database was password-protected, stored in the secured server of PACCI research center and accessible only to the research team.

## Results

Among the 108 HIV-2 clients in virologic success in 2012, 95 agreed to participate in the cohort. The median age at enrolment was 53 [47 – 60] years old, with gender parity and 56.8% of patients had secondary or higher education (Table 1). At enrolment, the median duration on ART was 9 [6 – 11] years, the median CD4 count was 319 [164 – 455] cells/mm$^3$ and all patients were receiving a PI based ART regimen.

After a median follow-up duration of 77 [75 – 78] months, 65 (68.4%) of the 95 participants remained in care, while 20 (21.1%) were LTFU and 10 (10.5%) were reported dead after a median follow-up duration of 40 [22 – 63] months. At the last follow-up visit 43 (45.3%) participants presented with WHO clinical stage 1-2/A-B, higher among patients remaining in care compared to those dead/LTFU (55.4% Vs 23.3%; p = 0.012). Advanced HIV disease was identified in 36 (37.9%) participants at their last follow-up visit, higher among those dead/LTFU compared to those remaining in care (60.0% vs 27.7%, p = 0.003) (Table 1). Of the 10 patients reported dead, four had a specified cause of death, including one heart attack, one heart failure, one renal failure and one cervical cancer.

The median CD4 count at enrollment in the cohort was 319 [164 – 455] cells/mm3, higher among those remaining in care compared to those dead/LTFU (351 Vs 222; p = 0.005). At the last follow-up visit, the median CD4 count was 518 [353 – 639] cells/mm$^3$, and there is no statistically significant difference in those remaining in care compared to those dead/LTFU (534 Vs 431; p = 0.09). Among the 65 participants remaining in care, the viral load was undetectable in 46 (70.8%) and detectable in 8 (12.3%) and those dead/LTFU did not have viral load recorded by the time of death or LTFU (Table 1).

The survival analysis revealed decreasing probability of remaining alive and in care over the time. At 24 months the probability of remaining alive and in care was 90% and decreased to 80.7% at 48 months and to 73% at 72 months (Fig 1).

## Discussion

The follow-up of treatment-experienced PLHIV-2 initially in virologic success, revealed that one-third of participants were dead or LTFU after 40 months, and almost two participants out of five presented with advanced HIV disease at their last follow-up visit. In addition, there was a decreasing probability of remaining alive and in care, around 73% after 72 months.

In our cohort, the proportion of patients who died or were LTFU was 30% five years following virologic success and enrollment in the cohort. This proportion was 27% in Burkina Faso in the same follow-up period and around 33% in a West African multi-country cohort of 1,825 PLHIV-2 after two years of follow-up [20,21]. In addition, a similar cohort of PLHIV-2 initially in virological failure, reported an attrition of 45%, after a median follow-up duration of five years. This result is higher than what we report here, but is concurring with the idea of important attrition among PLHIV-2 during the follow-up [22]. This important attrition rate among treatment experienced PLHIV-2 probably has the same drivers reported in PLHIV-1,

**Table 1. Follow-up characteristics of ART-experienced people living with HIV-2 five years following virologic success in West Africa (N = 95).**

| Characteristics | Total N = 95 | In care n = 65 | LTFU/dead n = 30 | p |
|---|---|---|---|---|
| **Age (Median; IQR), years** | 53 [45– 60] | 53 [47 – 59] | 53 [43 – 60] | |
| **Age class, years** | | | | 0.746 |
| <50 | 36 (37.9) | 23 (35.4) | 13 (43.3) | |
| ≥50 | 59 (62.1) | 42 (64.6) | 17 (56.7) | |
| **Gender** | | | | 0.944 |
| Male | 48 (50.5) | 33 (50.8) | 15 (50.0) | |
| Female | 47 (49.5) | 32 (49.2) | 15 (50.0) | |
| **Education level** | | | | 0.174 |
| No formal education/ Primary | 41 (43.2) | 25 (38.5) | 16 (53.3) | |
| Secondary/ High school | 54 (56.8) | 40 (61.5) | 14 (46.7) | |
| **Duration on ART (Median; IQR), years,** | 9 [6 – 11] | 10 [8–12] | 6 [4 – 8] | |
| **Class duration on ART, years** | | | | **<0.001** |
| 0–10 | 67 (70.5) | 41 (63.1) | 26 (86.7) | |
| ≥10 | 28 (29.5) | 24 (36.9) | 4 (13.3) | |
| **Follow-up duration in the cohort in months (median, IQR)** | 76 [53 – 78] | 77 [75 – 78] | 40 [22 – 63] | **<0.001** |
| **CD4 count at enrolment in the cohort (median, IQR), cells/mm³** | 319 [164 – 455] | 351 [199 – 479] | 222 [102 – 350] | **0.005** |
| **CD4 count at last visit (median, IQR), cells/mm³** | 518 [353 – 639] | 534 [383 – 662] | 431 [272 – 619] | **0.090** |
| **Class CD4 count at last visit, cells/mm³** | | | | 0.222 |
| >500 | 50 (52.6) | 38 (58.5) | 12 (40.0) | |
| [200–500] | 34 (35.8) | 21 (32.3) | 13 (43.3) | |
| <200 | 11 (11.6) | 6 (9.2) | 5 (16.7) | |
| **Last viral load** | | | | **<0.001** |
| Detectable | 8 (8.4) | 8 (12.3) | 0 (0.0) | |
| Undetectable | 46 (48.4) | 46 (70.8) | 0 (0.0) | |
| Not available | 41 (43.2) | 11 (16.9) | 30 (100.0) | |
| **WHO Clinical stage at last visit** | | | **0.012** | |
| 1-2/A-B | 43 (45.3) | 36 (55.4) | 7 (23.3) | |
| 3-4/C | 34 (35.8) | 18 (27.7) | 16 (53.4) | |
| Missing | 18 (18.9) | 11 (16.9) | 7 (23.3) | |
| **Last ART regimen** | | | | |
| 2 NRTI + PI/r | 95 (100.0) | 65 (100.0) | 30 (100.0) | |
| **Advanced HIV Disease at last visit (Y/N)** | | | | 0.003 |
| Yes | 36 (37.9) | 18 (27.7) | 18 (60.0) | |

such as male gender, advanced age and advanced HIV disease at treatment initiation, also, long distances to facility, transportation fees, long waiting time in the facility as well as stigma and discrimination [21,23,24]. Some of these were addressed by the implementation of differentiated service delivery model such as multi-month ART dispensation, community and family ART dispensation model, review of patients flow to enable the one-stop-shop model as well as training and sensitization of healthcare workers to reduce stigma and discrimination [25–28]. In addition, since 2019, national HIV treatment guidelines have recommended using the Tenofovir + Lamivudine + Dolutegravir (TLD) regimen as first-line treatment and switching to this treatment for patients already on antiretrovirals, with boosted protease inhibitors (PI/r) reserved as second-line. However, despite all these approaches and strategies, the long-term attrition among PLHIV-2 remained high, suggesting that additional efforts are needed to improve retention in this population. Improving holistic care to PLHIV-2 by

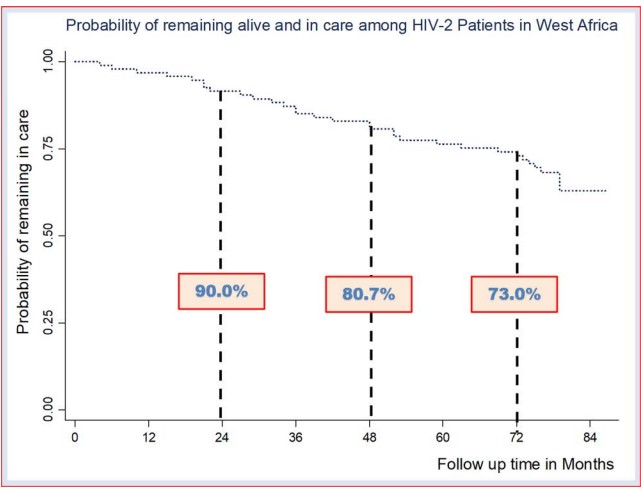

**Fig 1. Probability of remaining alive and in care among ART-experienced people living with HIV-2 five years following a virologic success.**

integrating the systematic screening and management of non-communicable diseases such as hypertension, diabetes, obesity, renal failure, cancers and mental health assessment as well as giving specific support to vulnerable PLHIV-2 could be part of the solution as suggested for PLHIV-1 [29,30].

Our study also revealed that nearly 40% of PLHIV-2 had advanced HIV disease, similar to 39.3% reported in an HIV-2 cohort in Guinea-Bissau [31]. These similar or even higher proportions in PLWH-2 could be explained by the fact that, before the recommendation of integrase inhibitors, treatment options for PLWH-2 were very limited due to the resistance of HIV-2 to certain ARVs. This situation exposed these patients to a higher risk of death in the event of failure of first-line ART [32]. Thus, despite progress in HIV treatment and prevention, the rate of advanced HIV disease still remains high in sub-Sahara African countries and this high rate can be explained by various forms of treatment interruptions as suggested in Botswana [33,34]. It is well known that people with advanced HIV disease are at high risk of dying from tuberculosis, severe bacterial infections or cryptococcal meningitis, in absence of early diagnosis and appropriate treatment [35]. Our results emphasize the critical need to make diagnosis and treatment of advanced HIV disease more accessible to PLHIV [36–38]. Indeed, since 2017, WHO published a specific guideline for the management of advanced HIV disease involving screening, treatment and prophylaxis of major AIDS-defining illnesses, prompt initiation of ART and intensified adherence support [39]. These include cotrimoxazole prophylaxis, the use of Xpert MTB/RIF, lateral flow lipoarabinomannan (LF-LAM) and isoniazid for the diagnosis and prevention of tuberculosis disease, as well as cryptococcal antigen (CrAg) screening and liposomal amphotericin B/ fluconazole for diagnosis and treatment of cryptococcosis [40,41].

In summary, innovative strategies such as tests that can qualitatively classified PLHIV regarding their immunological status (CD4 counts less than 200 cells/mm3 or CD4 counts equal or over 200 cells/mm3), the implementation of differentiated service delivery models, optimal management of opportunistic diseases where appropriate, and increased access to integrase inhibitor-based ARV regimens will help to effectively address attrition and highly advanced HIV disease among PLWH-2. Urging national AIDS control program and global health stakeholder to integrate advanced HIV disease package in the HIV program free of charge could help reducing mortality among all PLHIV including those living with HIV-2.

Some limitations should be acknowledged in our study, especially the small sample size that may limit the extrapolation of our results to the entire population of HIV-2 patients worldwide. Furthermore, viral load measurement was not introduced into treatment guidelines at the time of data collection. Consequently, viral load analyses only include those supported by the study at the start of the cohort and at the time of the 6-year analyses. In addition, our observation period was limited to December 2018, while WHO recommendation to use dolutegravir as preferred first line for all patients was implemented on a large scale in Côte d'Ivoire starting 2019. Since then, all HIV-2 patients on NRTI/PI have been switched to dolutegravir, including those on virologic failure needing to be switched to INSTI. This transition has probably improved the management of PLWH-2 in virological failure by offering an alternative second-line treatment that was previously limited. Despite these limitations, this study is among the few ones to describe long-term outcomes among PLHIV-2 in a context where data on follow-up of PLHIV-2 is very limited.

## Conclusion

Our study indicates that six years after being reported stable, one-third of PLHIV-2 were reported dead or LTFU. A high advanced HIV disease rate also reported among them call for the need to improve holistic care (enabling one-stop-shop model that allows early management of opportunistic infections and integrating non-communicable diseases services) for HIV-2 patients and fast-track the transition to DTG-based regimen.

## Acknowledgments

We would like to thank Jean Claude Azani for his assistance in managing and monitoring the data.

## Author contributions

**Conceptualization:** Didier K. Ekouevi, Boris Tchounga.

**Formal analysis:** Jean J. Koffi, Simon P. Boni, Boris Tchounga.

**Investigation:** Lionèle Mba.

**Methodology:** Boris Tchounga.

**Project administration:** Jean J. Koffi.

**Resources:** Lionèle Mba.

**Supervision:** Jean J. Koffi.

**Validation:** Didier K. Ekouevi, Boris Tchounga.

**Writing – original draft:** Jean J. Koffi, Simon P. Boni, Boris Tchounga.

**Writing – review & editing:** Alexandra Bitty-Anderson, Annick G. Tchabert, Frank Y. Toure, Patrick A. Coffie, André Inwoley, Serge Paul Eholie, Didier K. Ekouevi, Boris Tchounga.

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
