## [Decision Letter · Decision Letter 0]

29 Feb 2024

PONE-D-23-40207Mortality, loss to follow-up and advanced HIV disease following virologic success in West African HIV-2 patientsPLOS ONE

Dear Dr. Koffi,

Thank you for submitting your manuscript to PLOS ONE. After careful consideration, we feel that it has merit but does not fully meet PLOS ONE’s publication criteria as it currently stands. Therefore, we invite you to submit a revised version of the manuscript that addresses the points raised during the review process.

We look forward to receiving your revised manuscript.

Kind regards,

Joseph Fokam, Ph.D

Academic Editor

PLOS ONE

https://doi.org/10.1371/journal.pone.0236642

In your revision ensure you cite all your sources (including your own works), and quote or rephrase any duplicated text outside the methods section. Further consideration is dependent on these concerns being addressed.

4. One of the noted authors is a group [the EDIIMark-2 study group]. In addition to naming the author group, please list the individual authors and affiliations within this group in the acknowledgments section of your manuscript. Please also indicate clearly a lead author for this group along with a contact email address.

Reviewers' comments:

Reviewer's Responses to Questions

**Comments to the Author**

1. Is the manuscript technically sound, and do the data support the conclusions?

Reviewer #1: Yes

Reviewer #2: Yes

2. Has the statistical analysis been performed appropriately and rigorously? 

Reviewer #1: Yes

Reviewer #2: Yes

3. Have the authors made all data underlying the findings in their manuscript fully available?

Reviewer #1: Yes

Reviewer #2: Yes

4. Is the manuscript presented in an intelligible fashion and written in standard English?

Reviewer #1: No

Reviewer #2: Yes

5. Review Comments to the Author

Reviewer #1: This is a single-center observational study on HIV-2 in West Africa. While the disease is important, the information presented is not novel although as the authors mentioned, that this study joins the few studies to describe long-term outcomes (~5-¬¬6 years) among PLHIV-2. For e.g. it is not surprising that the patients remaining in care are more likely to have better outcomes with higher CD4 and a higher proportion in WHO clinical stage 1-2/A-B. Nevertheless, this study adds on to the sparse literature available. The manuscript will benefit with further English language editing if publishing in English.

Comments:

Abstract / Methods

It would be better to state that comparison categories included lost to follow up and dead. The current abstract appears to define loss to follow up = absent for more than 90 days and dead which is confusing.

Main manuscript / Under introduction

•Awkward language: “In addition, the 2019 WHO guidelines recommended Dolutegravir (DTG) in combination with a nucleoside reverse-transcriptase inhibitor (NRTI) backbone as the preferred first-line regimen for people initiating antiretroviral treatment (ART), thus without any difference between those living with HIV-1 and HIV-2 [14].” The reference to HIV-1 here is not required as well, as the authors are not comparing the treatment outcomes across HIV-1 and HIV-2.

Main manuscript / Under outcomes and variables

•While it would be fair for the authors to collapse mortality cases with lost to follow up if the numbers are small, I am concerned about the scientific basis stated that loss to follow up is a proxy of death.

•The preceding sentence mentioned that the authors categorized the participants into “3 modalities” and the next they collapsed 2 of them into 1. Modalities would not be the correct word to use here, but rather categories.

•It is not necessary to comment on the creation of a new variable. It would suffice to say that participants were categorized into 2 or 3 categories based on whether they are still on follow up or not.

Main manuscript / Data collection

•Awkward word use: Instead of abstraction, extraction would be a better word to use e.g. “A standardized questionnaire allowed extraction of data…”

Main manuscript / Results

•Was there any documented reason for the 13 HIV-2 clients to not participate in the survey? It would be interesting if yes, and this could be detailed.

•The median follow-up period for the last visit for the LTFU and dead group should be stated.

•The time point of HIV-2 viral load check should be mentioned and what the median and IQR are for those who are not considered controlled.

Main manuscript / Discussion

•The discussion does not really discuss about the findings of the research. For e.g. the research findings do not talk about opportunistic infections, but the discussion went into a paragraph of what would be helpful with opportunistic infections. In a cohort study wherein participants were initially all well-controlled, retainment in care would be more relevant rather than OI treatment.

•The finding that all participants are on boosted PI/NRTI regimen could have also been commented on or discussed. Depending on the HIV-2 viral load on those who are not controlled, there is a consideration wherein II based regimens would be more helpful.

Reviewer #2: The work is of great public health importance in the context of HIV-2 management in West Africa. The manuscript is well written in standard English, considering sound statistical methods.

For comments for the authors:

-Page 4, the spelling of house should be corrected. wrong "hose"; correct "house"

-On Data collection section, a standard data extraction form should have been used not a standardised questionnaire. Correct accordingly.

-Page 6, third paragraph and sentences 3 and 4 are confusing. I suggest you delete the 3rd sentence.

-Page 7, Table 1. Varaibles Age(median, IQR) years and Duration of ART (years) have 3 entries for only 2 categories. This needs to be reviewed.

Question: It was mentioned that all the PLHIV-2 at enrolment were of PI. During the follow-up period following the WHO recommendation to transition all patients to DTG, was this transition done? If yes, when and do you think it could affect the survival of PLHIV-2 or rate progress to advanced HIV disease? Consider addressing these elements in your work. Thanks

6. PLOS authors have the option to publish the peer review history of their article (what does this mean?). If published, this will include your full peer review and any attached files.

Reviewer #1: No

Reviewer #2: No

---

## [Author Response · Author response to Decision Letter 1]

29 Apr 2024

Authors: We would like to thank the editor for his comments. We have taken the additional requirements into account in the new version of the revised manuscript.

2. We noticed you have some minor occurrence of overlapping text with the following previous publication(s), which needs to be addressed: https://doi.org/10.1371/journal.pone.0236642

In your revision ensure you cite all your sources (including your own works), and quote or rephrase any duplicated text outside the methods section. Further consideration is dependent on these concerns being addressed.

Authors: We thank the editor for this comment, we reviewed the manuscript and removed all the overlaps with the above-mentioned paper.

Authors: We thank the reviewer for this comment, we provided a complete statement about data availability, we were able to make the data available on a repository and provided the link: https://figshare.com/s/a8b9fa7d5ba8a5968235

4. One of the noted authors is a group [the EDIIMark-2 study group]. In addition to naming the author group, please list the individual authors and affiliations within this group in the acknowledgments section of your manuscript. Please also indicate clearly a lead author for this group along with a contact email address.

Authors: We thank the reviewer for this comment, the Study group has been removed from the author list, since the list is very long and we could not join all the members to sign the related consent as per our organization policy.

Review Comments to the Author:

Reviewer #1: This is a single-center observational study on HIV-2 in West Africa. While the disease is important, the information presented is not novel although as the authors mentioned, that this study joins the few studies to describe long-term outcomes (~5-¬¬6 years) among PLHIV-2. For e.g. it is not surprising that the patients remaining in care are more likely to have better outcomes with higher CD4 and a higher proportion in WHO clinical stage 1-2/A-B. Nevertheless, this study adds on to the sparse literature available. The manuscript will benefit with further English language editing if publishing in English.

Authors: We thank the reviewer for this comment, we reviewed the manuscript to improve English and editing.

Comments:

Abstract / Methods

1• It would be better to state that comparison categories included lost to follow up and dead. The current abstract appears to define loss to follow up = absent for more than 90 days and dead which is confusing.

Authors: We would like to thank the reviewer for this comment. We clarified this confusion in the revised manuscript as follow: “Outcomes were described as follow: in care (known alive and present during the last ART follow up visit), loss to follow-up (absent for more than 90 days and not reported dead), and dead (reported dead with a date of event)".

Main manuscript / Under introduction

2• Awkward language: “In addition, the 2019 WHO guidelines recommended Dolutegravir (DTG) in combination with a nucleoside reverse-transcriptase inhibitor (NRTI) backbone as the preferred first-line regimen for people initiating antiretroviral treatment (ART), thus without any difference between those living with HIV-1 and HIV-2 [14].” The reference to HIV-1 here is not required as well, as the authors are not comparing the treatment outcomes across HIV-1 and HIV-2.

Authors: We thank the reviewer for this comment. We agree that there was no comparison between HIV-1 and HIV-2 planned in this analysis. Therefore, we removed the phrase “thus without any difference between those living with HIV-1 and HIV-2” in the revised version of the manuscript.

Main manuscript / Under outcomes and variables

3•While it would be fair for the authors to collapse mortality cases with lost to follow up if the numbers are small, I am concerned about the scientific basis stated that loss to follow up is a proxy of death.

Authors: We thank the reviewer for this comment. We decided to combine LTFU and death because there are papers in the literature that suggest that lost to follow-up is a proxy of death ("Loss to Follow-up: A Major Challenge to Successful Implementation of Prevention of Mother-to-Child Transmission of HIV-1 Programs in Sub-Saharan Africa). We included this paper as reference in the revised manuscript.

4• The preceding sentence mentioned that the authors categorized the participants into “3 modalities” and the next they collapsed 2 of them into 1. Modalities would not be the correct word to use here, but rather categories.

Authors: We thank the reviewer for this suggestion that has been taken into consideration by replacing “modalities” by “categories” in the revised version of the manuscript.

5• It is not necessary to comment on the creation of a new variable. It would suffice to say that participants were categorized into 2 or 3 categories based on whether they are still on follow up or not.

Authors: We thank the reviewer for this comment. We adjusted the sentence accordingly in the revised version of the manuscript.

Main manuscript / Data collection

6• Awkward word use: Instead of abstraction, extraction would be a better word to use e.g. “A standardized questionnaire allowed extraction of data…”

Authors: We thank the reviewer for this suggestion, we adjusted the text accordingly in the revised version of the manuscript.

Main manuscript / Results

7• Was there any documented reason for the 13 HIV-2 clients to not participate in the survey? It would be interesting if yes, and this could be detailed.

Authors: We thank the reviewer for this comment. Among this 13 people living with HIV-2 who did not participate to the cohort, height did not consent to participate and the remaining 5 have been transferred to facilities out of the town and not part of the study sites, making them not eligible.

8• The median follow-up period for the last visit for the LTFU and dead group should be stated.

Authors: We thank the reviewer for this comment. The median follow-up period for the last visit for the LTFU and dead group is (40 [22 - 63] months) and can be found in the results section of the manuscript, at the third line of paragraph 2.

9• The time point of HIV-2 viral load check should be mentioned and what the median and IQR are for those who are not considered controlled.

Authors: We thank the reviewer for giving an opportunity to update and improve the quality of our manuscript. In Cote d’Ivoire, HIV-2 Viral load was not routinely prescribed like CD4 count, because it was not supported by the HIV program. Thus, HIV-2 Viral load reported in our study was sponsored by the research project and done only at two-time point in 2012 and in 2018.

We recognized that median and IQR of viral load could be more informative. However, the lab results were not always given with the number of copies making it difficult to estimate the median number of copies for all the participants. We therefore made the choice to present the conclusion of the lab set as detectable or undetectable.

Main manuscript / Discussion

10• The discussion does not really discuss about the findings of the research. For e.g. the research findings do not talk about opportunistic infections, but the discussion went into a paragraph of what would be helpful with opportunistic infections. In a cohort study wherein participants were initially all well-controlled, retainment in care would be more relevant rather than OI treatment.

Authors: We thank the reviewer for this comment. We apologize for the confusion brought in the discussion when mentioning opportunistic infection. Our study did not focus on opportunistic infections as the reviewer rightly said, but rather explored attrition (LTFU and Death) and advanced HIV disease.

In the first graph we comment on the high attrition rate, giving the causes of this high rate and extrapolate on potential strategies to improve retention to care. In the second paragraph, we comment on the high rate of advanced HIV disease, and extrapolate on solutions that increase survival of patients. we adjusted the discussion section and tried to clarify that.

11• The finding that all participants are on boosted PI/NRTI regimen could have also been commented on or discussed. Depending on the HIV-2 viral load on those who are not controlled, there is a consideration wherein II based regimens would be more helpful.

Authors: We thank the reviewer for this comment. We added a comment in the discussion section reflecting the situation of participants receiving the recommended ART regiment for HIV-2 and experiencing an uncontrolled Viral load. Indeed, these patients should be switched on II. It is true that by 2018 the II were recommended only for the third lines, but with the roll out of DTG in 2019, all these patients were systematically switched to a DTG-based regimen. 

Reviewer #2:

1• Page 4, the spelling of house should be corrected. wrong "hose"; correct "house"

Authors: We would like to thank the reviewer for spotting this typing error. We corrected this misspelling in the update version of the manuscript.

2• On Data collection section, a standard data extraction form should have been used not a standardised questionnaire. Correct accordingly.

Authors: We thank the reviewer for this comment. The correct term is data extraction sheet. We have made a correction in the revised version of the manuscript.

3• Page 6, third paragraph and sentences 3 and 4 are confusing. I suggest you delete the 3rd sentence

Authors: We thank the reviewer for this comment. In the third sentence, we would like to show the proportions of the latest viral loads in our entire study population. However, we agree with the reviewer that it could be confused with the sentence that follows. In this revised version, we deleted this sentence as suggested by the reviewer.

4• Page 7, Table 1. Variables Age (median, IQR) years and Duration of ART (years) have 3 entries for only 2 categories. This needs to be reviewed.

Authors: We thank the reviewer for this valuable comment. The medians and categories of these variables overlapped and created confusion. We have distinguished them better in the new revised version of the manuscript.

5• Question: It was mentioned that all the PLHIV-2 at enrolment were of PI. During the follow-up period following the WHO recommendation to transition all patients to DTG, was this transition done? If yes, when and do you think it could affect the survival of PLHIV-2 or rate progress to advanced HIV disease? Consider addressing these elements in your work. Thanks

Authors: The WHO recommendation was implemented on a large scale in Côte d'Ivoire in 2019, after this study was carried out. Since then, all HIV-2 patients on NRTI/PI have been switched to dolutegravir. This transition has improved the management of PLWH-2 in virological failure by offering an alternative second-line treatment that was previously limited. We added a paragraph addressing the DTG transition in the discussion.

---

## [Decision Letter · Decision Letter 1]

8 Oct 2024

PONE-D-23-40207R1Mortality, loss to follow-up and advanced HIV disease following virologic success in West African HIV-2 patientsPLOS ONE

Dear Dr. Koffi,

Thank you for submitting your manuscript to PLOS ONE. After careful consideration, we feel that it has merit but does not fully meet PLOS ONE’s publication criteria as it currently stands. Therefore, we invite you to submit a revised version of the manuscript that addresses the points raised during the review process.Please ensure that your decision is justified on PLOS ONE’s publication criteria and not, for example, on novelty or perceived impact.

We look forward to receiving your revised manuscript.

Kind regards,

Zewdu Gashu Dememew, M.D, PhD

Academic Editor

PLOS ONE

Journal Requirements:

Additional Editor Comments:

Dear Authors,

Thank you very much for responding for comments from editor and reviewers.

There is a progress in this important article yet there are remaining issues to deal with.

Hope you will try your best to go through this important paper and come back with addressed suggestions and comments.

Good luck!

Reviewers' comments:

Reviewer's Responses to Questions

**Comments to the Author**

1. If the authors have adequately addressed your comments raised in a previous round of review and you feel that this manuscript is now acceptable for publication, you may indicate that here to bypass the “Comments to the Author” section, enter your conflict of interest statement in the “Confidential to Editor” section, and submit your "Accept" recommendation.

Reviewer #1: All comments have been addressed

2. Is the manuscript technically sound, and do the data support the conclusions?

Reviewer #1: Partly

3. Has the statistical analysis been performed appropriately and rigorously? 

Reviewer #1: Yes

4. Have the authors made all data underlying the findings in their manuscript fully available?

Reviewer #1: Yes

5. Is the manuscript presented in an intelligible fashion and written in standard English?

Reviewer #1: Yes

6. Review Comments to the Author

Reviewer #1: Thank you for the opportunity to review the revised manuscript.

Major

Discussion: Given that this cohort was only studied until 2018 and the national guidelines aligning with WHO guidelines occurred in 2019 – this alignment occurred post study. As such, the alignment of the national guidelines should not be included in “all these approaches and strategies, the long-term attrition” which the writing currently suggests. This is acknowledged as a limitation and I agree with the authors that a longer observation period would be useful to assess the effect of DTG transition.

Minor

Table 1 CD4 – brackets [200-500[ should be 200-500 ?

7. PLOS authors have the option to publish the peer review history of their article (what does this mean?). If published, this will include your full peer review and any attached files.

Reviewer #1: No

---

## [Author Response · Author response to Decision Letter 2]

22 Nov 2024

’This emphasizes the need to improve holistic care for HIV-2 patients.” The term holistic care is quite comprehensive for recommendation. Could the authors specify discrete recommendations that need to be done to reduce the advanced HIV diseases?

Authors: We would like to thank the reviewer for this comment. We agree that “holistic care” is definitely quite comprehensive to be part of a conclusion. We have updated the abstract by rewording it and adding practical take home messages. The conclusion has been updated accordingly by summarizing the key messages regarding to “holistic” care, discussed in the discussion section. Page 2, lines 52-54.

Introduction

The last paragraph should make sure that there is no published article in this regard- scarcity of evidence should strongly and boldly be described.

Examples:

Raugi DN, Ba S, Cisse O, Diallo K, Tamba IT, Ndour C, Badiane NM, Fortes L, Diallo MB, Faye D, Smith RA. Long-term experience and outcomes of programmatic antiretroviral therapy for human immunodeficiency virus type 2 infection in Senegal, West Africa. Clinical Infectious Diseases. 2021 Feb 1;72(3):369-78.

Also check why the term mortality started with capital letter here,” … Mortality, loss to follow-up (LTFU), advanced HIV disease…’’ Please check similar issues throughout the document.

Authors: We thank the reviewer for this comment. We have corrected this capitalization in the revised version of the manuscript. Page 3, line 81.

Methods

The following sentence is too long, and it is not clear. Please revise. Do the same throughout the document.

“A cohort of people living with HIV-2, identified as mono-infected and followed-up in HIV clinics participating in the International Epidemiological Database to Evaluate AIDS (IeDEA) West Africa, was initiated in January 2012.”

Authors: We thank the reviewer for this comment. In the revised version of the manuscript, we have split this long sentence into two sentences to make it easier to understand. Page 4, lines 85-87.

Follow up

Mix of tenses used, future and past. You may revise,” In case of missed appointment, a phone-base tracking procedure will take place to bring the patient back to care, and if not successful, home visits can be organized for the patients who agreed and provided localization of their house at enrolment. Blood sample was collected every six months from each patient to perform CD4 count measurements. For the purpose of this analysis, all the participants were…”

Authors: We thank the reviewer for this comment. We have standardized the (past) tense in the revised version of the manuscript. Page 4, line 96.

This is a vague sentence as well, “For the purpose of this analysis, all the participants were seen from October to December 2018 during their last appointment by the same clinician who conducted clinical examination and updated the disease clinical stage before collecting blood sample for viral load testing”

Authors: We recognize that this sentence is not very precise. We have reworded it in the revised manuscript to make it more understandable. Page 4, lines 99-103.

Data collections

 Again, mix of tenses,” A structured questionnaire allows extraction of data from patient’s medical files, and additional data were obtained during face-to-face interview with patients or abstracted in the facility databased.” Also check whether the word ‘abstracted’ is the right term to use here.

Authors: We've changed the sentence by harmonizing the tenses. The reviewer will find these changes in the revised version of the manuscript. Page 4, lines 108-110.

Check if the heading is “Outcomes and variables” or “Outcomes variables”. It seems only the outcome variables were operationalized. I strongly suggest to re-visit the PLOS ONE guidelines (https://journals.plos.org/plosone/s/submission-guidelines.#:~:text=The%20PLOS%20ONE%20article%20component%20must%20comply%20with%20the%20general)

Authors: The appropriate topic is 'Outcome variables', not 'Outcomes and variables'. We have made a correction in the revised version of the manuscript. Page 5, line 116.

Check why ‘Characteristic” started with capital letter here;” Table 1: Follow-up Characteristics of ART-experienced people living with HIV-2 five years following virologic success in West Africa (N=95)”

Authors: We have removed this major feature in the revised version of the manuscript. Page 7, line 172.

Gender parity-Table 1: how was the proportion of male and female became equal? Was it intentional? Please justify or check the data.

Authors: All patients with an undetectable viral load in 2012 were included in this study, regardless of gender. The equal proportion of men and women included in the analysis was by sheer coincidence. This result has already caught our attention during the data analysis phase, and even during the review by co-authors. After a second review, we confirmed it.

Now we are in 2024, and why could the authors not extend beyond 2018? Don’t you think the data is outdated?

Authors: The reviewer points out an interesting issue about this paper. First, after the success of the DLT transition into most of HIV programs, learning through knowledge building and sharing is still needed. Even our data seems outdated exclusively based on the data collection period, we think that they should contribute to more learning about the improving of PLHIV care and could serve programs managers as the outcomes of their efforts and progresses. In addition, as documenting in the limitation paragraph, at the discussion section, lessons learnt through this study suggest the potential benefic effect of Protease inhibitors (PI) - Dolutegravir transition on improving the management of PLWH-2 in virological failure by offering an alternative second-line treatment.

Based on these reasons, we would like to submit again the manuscript and call for its publication to nourish the science and contribute to the experience learned on the long-term outcomes of PLHIV under PI-based ART.

Is there any difference in terms of the outcome variables between the two countries? Is it possible to include this in the result and discuss the difference?

Authors: We thank the reviewer for this comment. For logistical reasons, the PLHIV-2 from Burkina Faso included in the 2012 cohort did not contribute to this analysis. This study only included the Côte d'Ivoire cohort. We have added this specificity to the method section to make it clearer. The revised version of the manuscript was also updated. Page 4, line 89.

It is expected to have a better outcome from treating HIV2 as compared to HIV1. Why was the advanced HIV diseases, LTFU and death all same or more than those patients infected with HIV1? Please try to justifying this issue further.

Authors: These similar or even higher proportions in PLWH-2 could be explained by the fact that prior to the recommendation of integrase inhibitors based ART, the therapeutic options for PLWH-2 were very limited. The PI-based ART regimen were indicated for first-line regimen, so ARVs resistances to this first-line made challenging the PLHIV-2 management. This situation exposed these patients to a higher risk of death if first-line ART failed. In addition, as with HIV-1, PLWH-2 presented at a late stage with severe immunosuppression, most commonly associated with opportunistic infections. In the revised version of the manuscript, we have modified the discussion to include this rationale. Page 9, lines 206-210.

In conclusion (abstract and at end of the discussion) please try to suggest specified interventions for the alarmingly high LTFU/death/Advanced HIV diseases:

-Management of comorbidities, counseling for adherence and retention, addressing adherence issue, considering a better regimens etc…

Authors: We thank the reviewer for this comment. Strategies to improve the management of PLWH-2 and to prevent advanced HIV disease according to WHO recommendations were expanded in the discussion. We have summarized some strategies in the revised version of the manuscript (including abstract). Page 10, lines 224-229.

Discussion

It is shallow and may be expanded a bit.

Potential point of discussion could be the difference in the outcomes between the two countries (Cote d’Ivoire and Burkin faso) or outcome similarity between HIV1 and HIV2.

Please try to trim/consolidate the limitation.

Authors: The data collected in this study only covered HIV clinics in Côte d'Ivoire, as reported earlier in the method section. We were therefore unable to make a comparison with data from Burkina Faso, which was not collected during the 2018 monitoring visit.

We have reduced the limitations section as suggested by the reviewer.

---

## [Editor Report · Decision Letter 2]

23 Dec 2024

Mortality, loss to follow-up and advanced HIV disease following virologic success in West African HIV-2 patients

PONE-D-23-40207R2

Dear Dr. Jean Jacques Koffi

We’re pleased to inform you that your manuscript has been judged scientifically suitable for publication and will be formally accepted for publication once it meets all outstanding technical requirements.

Kind regards,

Zewdu Gashu Dememew, M.D

Academic Editor

PLOS ONE

Dear Dr. Jean Jacques Koffi,

Thank you for coming up with the revised article. Major comments/suggestion well addressed.

Just consider the attached editorial suggestion prior to publication.

Best regards

Zewdu

Final comments to be fixed before consideration for publication

Abstract

Replace DTG with dolutegravir. Please look again in the manuscript—all abbreviations should be defined during their first instance if they are to be used again.

Introduction

“Data from the ANRS CO5 HIV-2 Cohort Study”. Same comment as above-- what is ANRS CO5 in, “ Data from the ANRS CO5 HIV-2 Cohort Study”? Also why capital letters?

Methods

‘’A cohort of people living with HIV-2 (PLHIV-2)…’’ only PLHIV-2 is enough. You have already defined.

Check again if it is important to start these phrases with capital letters, “ As per the National Guidelines for PLHIV, clinically…”

Check if the following long sentence could be revised to two sentences.

’’ Based on their follow up status, participants were categorized as in care (still in the follow-up and receiving ART, including being transferred out), dead and LTFU, and the last two were subsequently

combined in one category in the analyses as they both constitute a negative outcome and it is always challenging to know among the LTFU the proportion of death (19).”

May be replaced with;

‘’ Based on their follow up status, participants were categorized as in care (still in the follow-up and receiving ART, including being transferred out), dead and LTFU. The last two were subsequently

combined in one category in the analyses as they both constitute a negative outcome, and it is

always challenging to the proportion of death know among the LTFU (19).’’

Replace “Chi2-square or Fisher’s exact tests ..” with “Chi-square (X2) or Fisher’s exact tests ….”

Replace “Prior to the enrolment in the cohort, each participant received detailed information about the study and signed an informed consent form.”

With

“Prior to the enrolment in the cohort, detailed information about the study was given to each study participants and a signed informed consent was obtained .”

What is “PACCI”? Please define.

Check if “PI based ART regimen…” Could be written as, “ Protease inhibitors (PI) based ART regimen…”

Check if the sentence, “At the last follow-up visit, the median CD4 count was 518 [353 – 639] cells/mm3 not different in those remaining in care compared to those dead/LTFU (534 Vs 431; p=0.09).”

could better be written as,

“At the last follow-up visit, the median CD4 count was 518 [353 – 639] cells/mm3, and there is no statistically significant difference in those remaining in care compared to those dead/LTFU (534 Vs 431; p=0.09).”

Discussion

“Moreover, since 2019, national HIV treatment guidelines aligned on WHO guidelines

recommended Tenofovir plus Lamivudine plus dolutegravir (TLD) regimen as first-line and the

transition of all ART-experienced clients, while the use of boosted protease inhibitors (PI/r) as

second-line.” This is not easy to understand. Please simplify or written in a simple sentence for easy understanding.

Conclusions

Which one is right?

“..five years after an initial virologic success…” in the abstract conclusion

or

“Our study indicates that six years after being reported stable,…” in the final conclusion?

---

## [Editor Report · Acceptance letter]

PONE-D-23-40207R2

PLOS ONE

Dear Dr. Koffi,

I'm pleased to inform you that your manuscript has been deemed suitable for publication in PLOS ONE. Congratulations! Your manuscript is now being handed over to our production team.

Kind regards,

on behalf of

Dr. Zewdu Gashu Dememew

Academic Editor

PLOS ONE